

# Compositions of the major ions, variations in their sources, and a risk assessment of the Qingshuijiang River Basin in Southwest China: a 10-year comparison of hydrochemical measurements

Jiemei Lv[1], Tianhao Yang[2] and Yanling An[1]

[1] The College of Resources and Environmental Engineering, Guizhou Institute of Technology, Guiyang, Guizhou, China
[2] School of Public Health, the Key Laboratory of Environmental Pollution Monitoring and Disease Control, Ministry of Education, Guizhou Medical University, Guiyang, Guizhou, China

Corresponding author
Yanling An, anyanling@git.edu.cn

## ABSTRACT

Rivers in karst areas face increased risks from persistent growth in human activity that leads to changes in water chemistry and threatens the water environment. In this study, principal component analysis (PCA), ion ratio measurements, and other methods were used to study the water chemistry of the Qingshuijiang River Basin over the past 10 years. The results showed that the main ions in the river were $Ca^{2+}$ and $HCO_3^-$, with a cation order of $Ca^{2+}$ (mean: 0.93 mmol/L) > $Mg^{2+}$ (mean: 0.51 mmol/L) > $Na^+$ (mean: 0.30 mmol/L) > $K^+$ (mean: 0.06 mmol/L) and $HCO_3^-$ (mean: 2.00 mmol/L) > $SO_4^{2-}$ (mean: 0.49 mmol/L) > $Cl^-$ (mean: 0.15 mmol/L) > $NO_3^-$ (mean: 0.096 mmol/L) > $F^-$ (mean : 0.012 mmol/L). In the past 10 years, the concentration of major ions in the river water in the basin has increased significantly. The weathering input of rock (mainly upstream carbonate) was the main source of $Mg^{2+}$, $Ca^{2+}$, and $HCO_3^-$, though sulfuric acid was also involved in this process. While $K^+$ and $Na^+$ were affected by the combination of human activity and the weathering input of silicate rock in the middle and lower reaches of the river, human activity was the main source of $SO_4^{2-}$, $NO_3^-$, and $F^-$ ions. Irrigation water quality and health risks were evaluated by calculating the sodium adsorption ratio (SAR), soluble sodium percentage (Na%), residual sodium carbonate (RSC), and hazard quotient (HQ). The findings indicated that the river water was generally safe for irrigation and drinking, and the health risks were gradually reduced over time. However, long-term monitoring of the river basin is still essential, especially for the risk of excessive $F^-$ in a few tributaries in the basin.

# INTRODUCTION

Rivers are vital sources of water for agriculture, industry, and human consumption (*Hayward et al., 2022*). Rapid urbanization and increased population density have led to the discharge of human activities (urban sewage, industrial wastewater, and agricultural

wastewater) into rivers (*Hua et al., 2020*; *Li et al., 2019*). This process has changed the chemical evolution of river water, resulting in water shortages and water quality deterioration, endangering human health (*Han & Xu, 2022*; *Yao et al., 2015*). In general, the substances in rivers are derived from both natural processes (such as atmospheric rainfall and rock weathering) and human activity (*Qin et al., 2018*; *Zheng et al., 2022*). However, human activity has gradually become the main factor affecting the chemical composition of the river water environment. In particular, the direct input of $F^-$, $Cl^-$, $SO_4^{2-}$, $NO_3^-$ and other elements from agricultural and urban areas significantly changes the hydrochemical characteristics of rivers (*Jehan et al., 2019*; *Sheng et al., 2023*; *Zheng et al., 2022*). Due to differences in climate, lithology, and human activity, river water chemistry and their control mechanisms differ by geography (*Hua et al., 2020*; *Wang et al., 2022a*; *Yu et al., 2021*). Long-term and seasonal research on river water chemistry can intuitively reflect the evolution and effects of multiple factors on water chemistry (*Feng & Yang, 2022*).

The main ions in river water ($Na^+$, $K^+$, $Mg^{2+}$, $Ca^{2+}$, $F^-$, $Cl^-$, $NO_3^-$, $SO_4^{2-}$, and $HCO_3^-$) are important components of the river dissolution load, and their geochemical behavior can further determine the main contribution source and migration process of river pollutants (*Li et al., 2022*). Recent studies have indicated that ion ratio, principal component analysis (PCA), and correlation analysis (CA) are the most common methods for analyzing the main ion sources in river water (*Han et al., 2022*; *Long & Luo, 2020*; *Mingyue et al., 2024*; *Zhang et al., 2022*). Statistical methods such as PCA and CA can be used to explore the common source of river solutes, and the ion ratio can help prevent the potential dilution effect and reflect the mixing process of the source (*Gaillardet et al., 1999*; *Szynkiewicz et al., 2011*). Therefore, a multivariate statistical analysis of rivers can more accurately identify the contribution sources of each ion. Increased human activity leads to an increase in ion concentrations in the water environment, leading to serious health and environmental risks (*Wu et al., 2017*). Water pollutants mainly enter the human body through drinking water and skin contact, endangering human health (*Liu et al., 2021a*, *2021b*). Rivers are important water sources for agricultural irrigation, and the water quality of the river directly affects the growth of crops around the basin (*Asare-Donkor, Ofosu & Adimado, 2018*). Therefore, the health risk assessment (HRA) and irrigation water quality measurements of rivers are of great significance to the health of the local population. Recent studies focus on the risk assessment method calculated by hazard quotient (HQ) to assess the risk of direct exposure to residents, and the HQ index is also widely used to evaluate the impact of water pollution on human health (*Gao et al., 2021b*; *Wang et al., 2022b*). The evaluation of irrigation water quality using the sodium adsorption ratio (SAR) and soluble sodium percentage (Na%) reflects the degree of alkali (sodium) damage in the irrigation water (*Han & Xu, 2022*).

Southwest China is a typical karst area (*Jiang, Lian & Qin, 2014*). Due to unique geological conditions and strong karstification, karst ecosystems, especially river systems, are extremely sensitive and fragile (*Gutiérrez et al., 2014*). This study assessed seasonal changes and time differences in the river hydrochemistry of karst areas to better explore the effects of human activity on river hydrochemical characteristics, the weathering

process, and the health risks of rivers in karst areas. The river ions in the Qingshuijiang River Basin were systematically studied during the wet and dry seasons of 2013/2014 and 2023, aiming to (1) analyze the spatial and temporal variation of the main ions in the Qingshuijiang River Basin, (2) explore the sources and changes of the main ions in the basin, and (3) assess the water quality changes and potential health risks in the basin.

## MATERIALS AND METHODS

### Study area

The Qingshuijiang River Basin is located in the eastern and central part of Guizhou Province (southwestern karst area; Fig. 1A), between 105°15′–109°50′ east longitude and 26°10′–27°15′ north latitude. It is one of the important tributaries of the Dongting River System in the upper reaches of the Yuanjiang River Basin in the Yangtze River Basin. In the upper portions of the basin, Duyun, Fuquan, and Kaili have experienced rapid industrial development. The enterprises producing phosphorus and fluoride in the basin are mainly concentrated upstream in Fuquan City, which has built a large-scale phosphate rock and phosphorus chemical base. The soil in the Qingshuijiang River Basin mainly includes yellow soil, red soil, yellow-red soil, and red paddy soil. The forest resources in the area are abundant, and the forest coverage rate of the basin is about 50%. The rock distribution in the upper reaches of the basin is mainly dolomite, limestone, sand shale, clastic rock, and marl. The middle and lower reaches consist of mainly siliceous rock, slate, metamorphic sandstone, metamorphic tuff, and sedimentary tuff. There is abundant rainfall in the basin, averaging 1,050−1,500 mm annually, and the average annual temperature is 14−18 °C.

### Sampling and measurements

Based on the rock distribution characteristics and urban land distribution in the basin, surface water samples from the whole basin were collected in August 2013 (wet season), January 2014 (dry season), February 2023 (dry season) and September 2023 (wet season). A total of 164 samples, including 68 mainstream samples (G1–G17) and 96 tributary samples (Z1–Z24), were collected at a water depth of about 15 cm using clean, high-density polyethylene bottles (Fig. 1B). The water samples were immediately filtered through a 0.45 µm Millipore filter membrane in the field after collection. The water samples used for the cation analysis ($Na^+$, $K^+$, $Ca^{2+}$, and $Mg^{2+}$) and for the anion analysis ($F^-$, $Cl^-$, $NO_3^-$, and $SO_4^{2-}$) were sealed and stored away from light. The pH, conductivity (EC), water temperature (T), and dissolved oxygen (DO) of the water body were measured on site using a WTW portable multi-parameter tester. $HCO_3^-$ was titrated by 0.025 mol·L$^{-1}$ HCl on site, and each water sample was titrated three times to ensure that the volume error of HCl used each time was within 0.1 ml. $Na^+$, $K^+$, $Ca^{2+}$, $Mg^{2+}$, $Cl^-$, $NO_3^-$, $F^-$, and $SO_4^{2-}$ were determined by ion chromatography (DIONEX, ICS-1100, IonPac AG-19 anion column, IonPac CS-12A cation column). The test accuracy of parallel samples is better than ±5%.
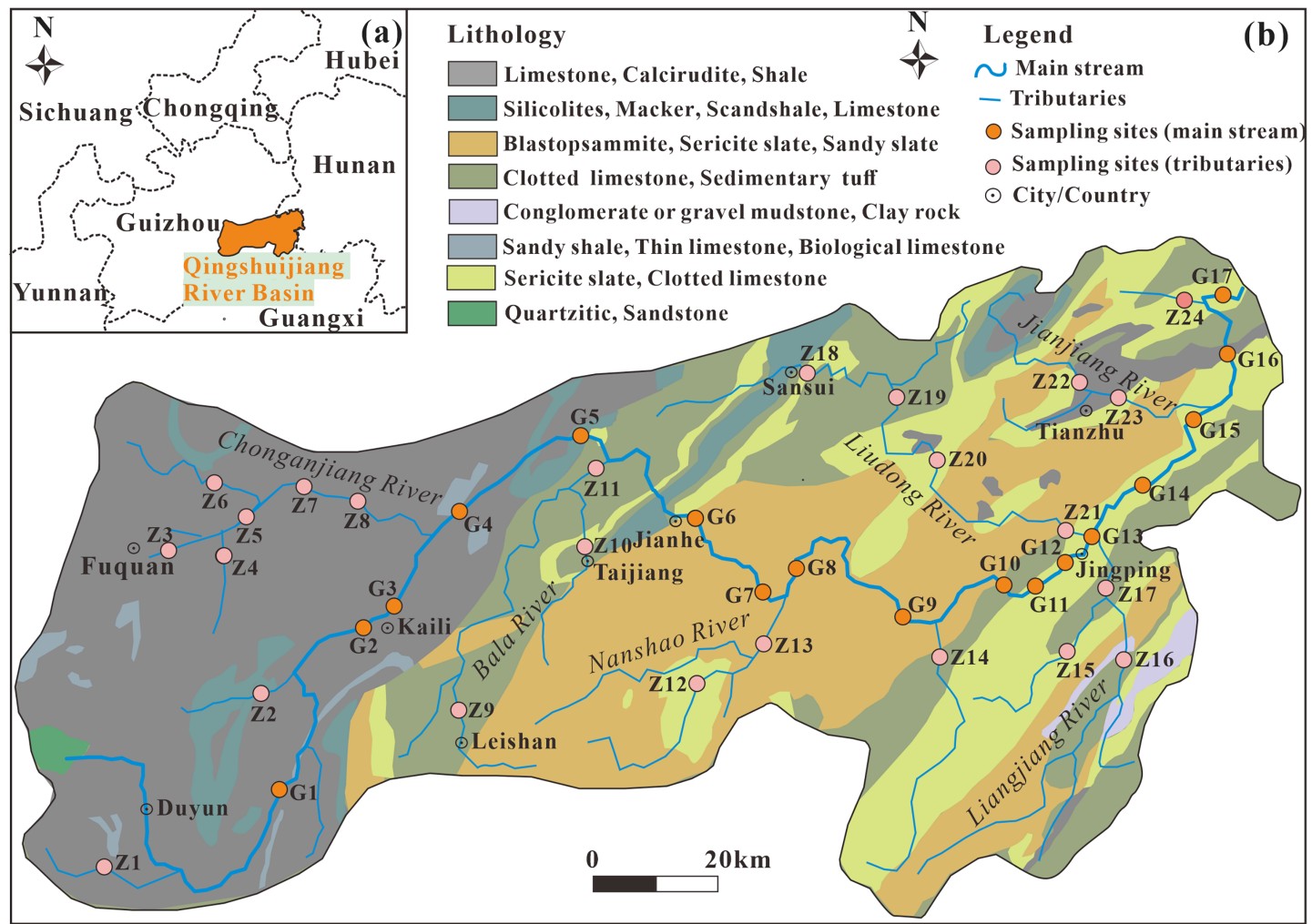

**Figure 1 Location of the Qingshuijiang River Basin in Guizhou, Southwest China (A); sampling lithology maps of the mainstream and tributaries of the Qingshuijiang River Basin (B).** Data source: Geocloud (https://geocloud.cgs.gov.cn).

## Assessment method

### Irrigation water quality assessment

Irrigation water quality was evaluated by calculating the sodium adsorption ratio (SAR), soluble sodium percentage (Na%), and residual sodium carbonate (RSC). Higher SAR values indicate stronger adsorption of sodium ions by the soil in the basin, making it more difficult for the vegetation roots to absorb water (*Wang et al., 2022b*). The higher the Na% value, the worse the permeability of the soil, affecting the growth of vegetation. Different irrigation water salinity and alkalinity levels also affect soil quality attributes, thereby changing farmland yield. Therefore, SAR, Na%, and RSC were calculated using the concentration (meq/L) of ions ($Na^+$, $K^+$, $Mg^{2+}$, $Ca^{2+}$, $HCO_3^-$, and $CO_3^{2-}$) in river water to comprehensively evaluate the saline-alkali hazard of irrigation water (*Asare-Donkor, Ofosu & Adimado, 2018*). The main calculation formulas are as follows:

$$SAR = \sqrt{2} \times Na^+ / (Ca^{2+} + Mg^{2+})^{1/2} \tag{1}$$

$$Na\% = Na^+ / (Ca^{2+} + Mg^{2+} + Na^+ + K^+) \times 100\% \tag{2}$$

$$RSC = (HCO_3^- - CO_3^{2-}) - (Ca^{2+} + Mg^{2+}) \tag{3}$$

### Health risk assessment

Water pollutants in river water enter the human body mainly through both drinking water and skin contact, and unsafe drinking water intake has been shown to be the main way to threaten human health (*Adimalla & Li, 2019*; *Adimalla, Qian & Nandan, 2020*). Long-term exposure to high concentrations of nitrate and fluoride may increase the risk of disease and other negative health effects (*Xia et al., 2021b*). Recent research has shown that $NO_3^-$ and $F^-$, as non-carcinogenic pollutants, are often used to assess the non-carcinogenic health risks of river water to the population (*Han & Xu, 2022*; *Wang et al., 2017*; *Zeng, Han & Yang, 2020*), though the hazard quotient (HQ) is the most commonly used metric for evaluating non-carcinogenic health risks. Its calculation method follows in Eqs. (4) and (5):

$$ADD_{ingestion} = C \times IR \times EF \times ED / (BW \times AT) \tag{4}$$

where the $ADD_{ingestion}$ is the ingestion intake of daily doses, C is the concentrations of ions (mg/L), IR is the rate of daily ingestion (0.6 L/day for children, 1 L/day for adults), EF is the exposure frequency (365 days/year for both adults and children), ED is the exposure duration (25 years for adults and 12 for children), BW is the body weight (16 kg for children and 56 kg for adults), and AT is the average time (4,380 days for children and 10,950 days for adults; (*Qasemi et al., 2019*; *Wu & Sun, 2016*)).

$$HQ = ADD_{ingestion} / RfD_{ingestion} \tag{5}$$

where $DfR_{ingestion}$ is the reference dose of different ions (0.04 and 1.60 ppm/day for $F^-$ and $NO_3^-$, respectively; (*Li et al., 2016*)). When HQ < 1, the human health risk caused by pollutants is permissible; when HQ > 1, non-carcinogenic effects should be considered.

## Data analysis

In this study, ion ratio, principal component analysis, correlation analysis, and piper three-line diagram were used to analyze the compositions, source, and changes of the main ions in the Qingshuijiang River Basin. Statistics software package SPSS 25.0 and Origin 2022 were used for data analysis and visual representation of the data.

## RESULTS

### Water quality parameters of the Qingshuijiang River

The minimum, maximum, mean, and standard deviation of water quality parameters and main ion concentrations in the Qingshuijiang River Basin in 2013/2014 and 2023 are summarized in Table 1. The water temperature of the Qingshuijiang River Basin ranged from 7.90 °C to 30.60 °C (average: 18.58 °C), and the pH value ranged from 7.07 to 9.90
Lv et al.
2024
10.7717/peerj.18284

**Table 1 Values of major ions and some hydrogeochemical parameters in the Qingshuijiang River Basin.**

| Sampling time | | T | EC | pH | DO | Na$^+$ | K$^+$ | Mg$^{2+}$ | Ca$^{2+}$ | F$^-$ | Cl$^-$ | NO$_3^-$ | SO$_4^{2-}$ | HCO$_3^-$ | TDS | SAR | Na% | RSC |
|---|---|---|---|---|---|---|---|---|---|---|---|---|---|---|---|---|---|---|
| | | °C | µS/cm | | mg/L | mmol/L | | | | | | | | | mg/L | | | |
| 2013/8[a] | Min | 21.70 | — | 7.19 | 3.30 | 0.04 | 0.02 | 0.04 | 0.09 | 0.002 | 0.03 | 0.001 | 0.04 | 0.37 | 33.82 | 0.05 | 2.28 | 0.24 |
| | Max | 30.60 | — | 9.90 | 12.30 | 0.35 | 0.09 | 1.12 | 1.62 | 0.083 | 0.18 | 0.122 | 1.05 | 4.61 | 500.64 | 0.60 | 50.00 | 2.50 |
| | Mean | 25.98 | — | 8.05 | 7.05 | 0.18 | 0.04 | 0.44 | 0.76 | 0.014 | 0.10 | 0.022 | 0.35 | 1.91 | 201.51 | 0.29 | 18.08 | 0.71 |
| | SD | 2.06 | — | 0.64 | 1.93 | 0.07 | 0.02 | 0.38 | 0.46 | 0.018 | 0.04 | 0.027 | 0.29 | 1.17 | 132.59 | 0.15 | 11.88 | 0.44 |
| 2014/1[a] | Min | 8.00 | 37.7 | 7.07 | 5.62 | 0.04 | 0.01 | 0.03 | 0.07 | 0.001 | 0.02 | 0.002 | 0.05 | 0.18 | 21.46 | 0.04 | 1.97 | −0.34 |
| | Max | 16.00 | 1,186 | 9.07 | 12.84 | 1.08 | 0.08 | 1.32 | 1.75 | 0.174 | 0.22 | 2.229 | 2.43 | 4.09 | 761.70 | 0.87 | 54.17 | 1.62 |
| | Mean | 11.33 | 302.3 | 7.86 | 9.47 | 0.24 | 0.04 | 0.50 | 0.88 | 0.017 | 0.10 | 0.178 | 0.44 | 1.62 | 210.10 | 0.32 | 18.11 | 0.25 |
| | SD | 2.65 | 211.0 | 0.41 | 1.72 | 0.18 | 0.01 | 0.38 | 0.51 | 0.028 | 0.04 | 0.377 | 0.43 | 1.00 | 161.90 | 0.17 | 11.83 | 0.33 |
| 2023/2 | Min | 7.90 | 58.6 | 7.86 | 6.33 | 0.06 | 0.01 | 0.04 | 0.09 | 0.004 | 0.03 | 0.013 | 0.06 | 0.26 | 29.74 | 0.05 | 2.21 | −2.55 |
| | Max | 14.60 | 1,317 | 8.55 | 12.50 | 2.51 | 0.39 | 1.38 | 4.75 | 0.072 | 1.01 | 0.493 | 4.88 | 5.01 | 1,137.74 | 1.81 | 51.82 | 2.56 |
| | Mean | 11.13 | 395.7 | 8.26 | 10.36 | 0.46 | 0.08 | 0.62 | 1.13 | 0.010 | 0.24 | 0.115 | 0.69 | 2.36 | 299.68 | 0.48 | 21.23 | 0.62 |
| | SD | 1.52 | 276.7 | 0.17 | 1.07 | 0.52 | 0.08 | 0.42 | 0.96 | 0.012 | 0.21 | 0.098 | 1.04 | 1.19 | 250.29 | 0.32 | 10.58 | 0.84 |
| 2023/9 | Min | 21.80 | 23.1 | 7.65 | 2.43 | 0.07 | 0.01 | 0.04 | 0.09 | 0.005 | 0.02 | 0.002 | 0.04 | 0.34 | 31.94 | 0.07 | 2.89 | −1.23 |
| | Max | 29.70 | 1,001 | 9.65 | 17.86 | 1.38 | 0.39 | 1.26 | 3.60 | 0.030 | 0.40 | 0.381 | 3.19 | 4.76 | 856.68 | 1.05 | 50.90 | 2.40 |
| | Mean | 25.86 | 305.7 | 8.47 | 8.01 | 0.31 | 0.07 | 0.48 | 0.97 | 0.007 | 0.15 | 0.069 | 0.49 | 2.11 | 245.98 | 0.39 | 20.10 | 0.66 |
| | SD | 1.87 | 198.5 | 0.44 | 2.55 | 0.25 | 0.06 | 0.32 | 0.68 | 0.005 | 0.08 | 0.064 | 0.60 | 1.08 | 173.02 | 0.18 | 10.27 | 0.55 |
| Four periods | Min | 18.58 | 334.6 | 8.16 | 8.72 | 0.30 | 0.06 | 0.51 | 0.94 | 0.012 | 0.15 | 0.096 | 0.49 | 2.02 | 240.71 | 0.04 | 1.97 | −2.55 |
| | Max | 30.60 | 1,317 | 9.90 | 17.86 | 2.51 | 0.39 | 1.38 | 4.75 | 0.174 | 1.01 | 2.229 | 4.88 | 5.01 | 1,137.74 | 1.81 | 54.17 | 2.56 |
| | Mean | 18.58 | 334.6 | 8.16 | 8.72 | 0.30 | 0.06 | 0.51 | 0.94 | 0.012 | 0.15 | 0.096 | 0.49 | 2.00 | 239.32 | 0.37 | 19.38 | 0.56 |
| | SD | 2.03 | 228.7 | 0.42 | 1.82 | 0.26 | 0.04 | 0.38 | 0.65 | 0.016 | 0.09 | 0.142 | 0.59 | 1.14 | 166.03 | 0.22 | 11.14 | 0.60 |
| Chinese guideline[b] | | — | — | 6.5–8.5 | — | — | — | — | — | 0.050 | 7.05 | 1.430 | 2.60 | — | — | — | — | — |
| WHO guideline[b] | | — | — | 6.5–8.5 | — | — | — | — | — | 0.080 | 7.05 | 3.570 | 2.60 | — | — | — | — | — |

Notes:
[a] The concentrations in the Qingshuijiang River Basin in August 2013 and January 2014 were derived from *Lü et al. (2018)*.
[b] The unit of related values in Chinese guideline and WHO guideline are converted to mmol/L.

(average: 8.16), which is neutral to weak alkaline. Generally, the EC and total dissolved solids (TDS) of a body of water can be used to reflect its ionic strength (*Zhou et al., 2016*). The EC of the Qingshuijiang River Basin ranged from 23.1 to 1,317 µS/cm (average: 334.6 µS/cm), and the TDS ranged from 24.35 to 1,031.87 mg/L (average: 239.32 mg/L). The total cationic charges (TZ$^+$ = Na$^+$ + K$^+$ + 2Mg$^{2+}$ + 2Ca$^{2+}$) averaged 3.24 meq/L, and the anionic charges (TZ$^-$ = F$^-$ + Cl$^-$ + NO$_3^-$ + 2SO$_4^{2-}$ + HCO$_3^-$) also averaged 3.24 meq/L. The normalized inorganic charge balance (NICB, [TZ$^-$−TZ$^+$]/TZ$^-$) of the samples was within ±15% except for the sampling point Z3 (heavily polluted tributaries) in the dry season of 2014, and 80% of the samples were within ±10%.

Compared with the dry season of 2014, the EC and TDS of the dry season of 2023 were significantly higher, indicating that the ion content in the river water has increased over the past decade. Moreover, the TDS in the dry season of 2023 was significantly higher than the TDS in the wet season of 2023. Over the past decade, Na$^+$, K$^+$, Cl$^-$, and SO$_4^{2-}$ in the Qingshuijiang River Basin have increased by 83.33%, 87.50%, 95.00%, and 49.37%,

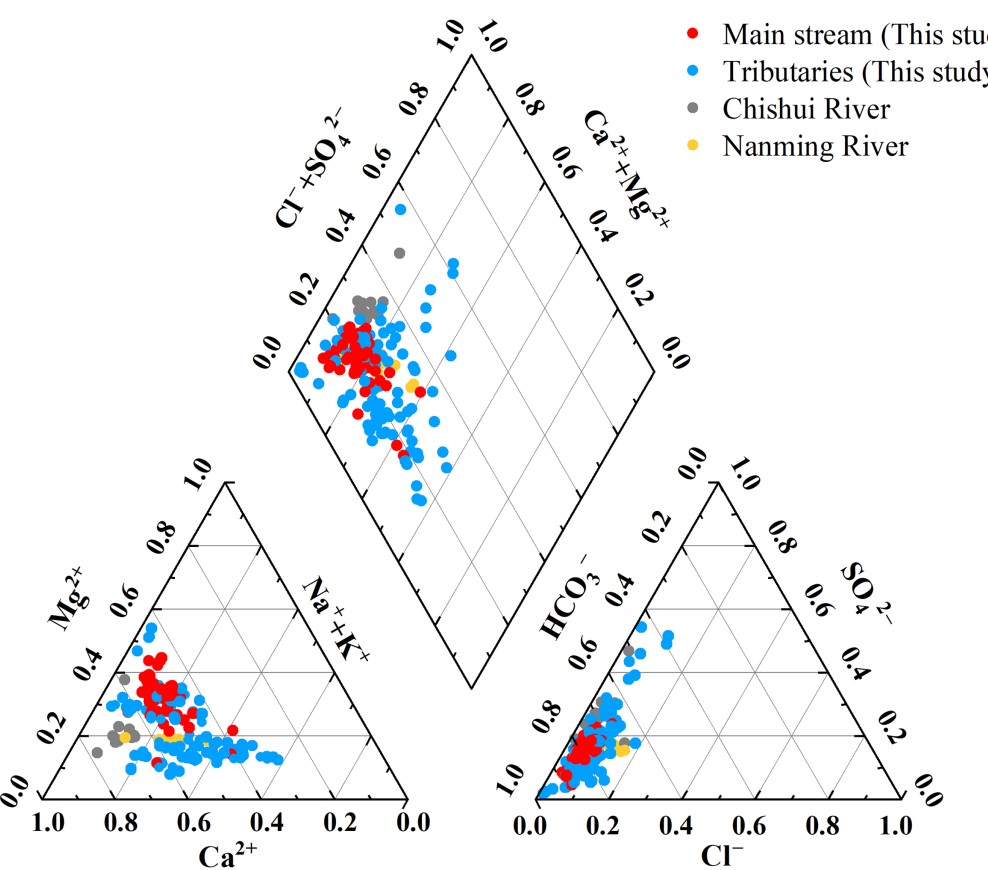

**Figure 2** **Piper trilinear diagram of the major ion geochemistry of the Qingshuijiang River and other karst rivers.** Data sources: (*Ge et al., 2021*) for Chishui River, and *Han & Xu (2022)* for Nanming River.

respectively. In addition, $Ca^{2+}$ and $Mg^{2+}$ in the river water have increased slightly, while $F^-$ and $NO_3^-$ have decreased, especially $F^-$, which has decreased by 46.16%.

## Hydrochemical characteristics of the Qingshuijiang River

The results showed that the main ions in the Qingshuijiang River were $Ca^{2+}$ and $HCO_3^-$, with a cation order of $Ca^{2+}$ (0.93 mmol/L) > $Mg^{2+}$ (0.51 mmol/L) > $Na^+$ (0.30 mmol/L) > $K^+$ (0.06 mmol/L) and $HCO_3^-$ (2.00 mmol/L) > $SO_4^{2-}$ (0.49 mmol/L) > $Cl^-$ (0.15 mmol/L) > $NO_3^-$ (0.096 mmol/L) > $F^-$ (0.012 mmol/L). The results also showed that the main anions of Qingshuijiang River were $HCO_3^-$ and $SO_4^{2-}$, and the main cations were $Ca^{2+}$ and $Mg^{2+}$, which is consistent with the composition characteristics of the main ions in karst rivers (Chishui River and Nanming River) in Guizhou (*Ge et al., 2021*; *Han & Xu, 2022*). A Piper diagram was then used to better understand the hydrochemical characteristics of the Qingshuijiang River Basin (*Xia et al., 2021a*). As shown in Fig. 2, the main cations, $Ca^{2+}$ and $Mg^{2+}$, accounted for 50.88% and 26.19% of the total cations, respectively. The main anion, $HCO_3^-$, accounted for 77.47% of the total anions, followed by $SO_4^{2-}$, which accounted for 16.31% of the total anions. $Ca^{2+}$-$HCO_3^-$ type water was the main hydrochemical water type of the Qingshuijiang River. In the upstream tributary area,

which has abundant human activity, the hydrochemical water type changed from $Ca^{2+}$-$HCO_3^-$ type water to $Ca^{2+}$-$SO_4^{2-}$ type water. The middle and lower reaches of the tributary area are weathered by silicate rocks, and the hydrochemical water type changed from $Ca^{2+}$-$HCO_3^-$ water type to $Na^+$-$K^+$-$HCO_3^-$ water type (*Fantong et al., 2020*; *Mufur et al., 2021*).

## DISCUSSION

### PCA and CA analysis of solute sources

To further analyze the source of the main ions in the Qingshuijiang River, a principal component analysis (PCA) was used to divide the main ions in the river water into several different components, and a correlation analysis (CA) was used to further explore the relationship between ions in the same component and identify whether there was a common source (*Gao et al., 2021a*; *Zhou et al., 2023*; *Yan et al., 2023*). The Kaiser-Meyer-Olkin (KMO) test result was 0.776, which was used for factor analysis, and the principal components (PCs) were divided into three parts: PC1: 64.41% variance, PC2: 15.45% variance, and PC3: 13.25% variance (Table 2). Additionally, all the eigenvalues were greater than 1, and the cumulative contribution rate of the three PCs reached 93.11%. PC1 exhibited high positive loadings (>0.50) of $Cl^-$, $SO_4^{2-}$, $Na^+$, $K^+$, and $Ca^{2+}$; PC2 exhibited high positive loadings (>0.50) of $Mg^{2+}$, $Ca^{2+}$, and $HCO_3^-$; and PC3 exhibited high positive loadings (>0.50) of $F^-$ and $NO_3^-$.

The upstream Qingshuijiang River Basin is a karst carbonate geological area, and the downstream Qingshuijiang River Basin is a silicate geological area. PC1 represented a variety of sources of ions. The weathering of silicate rocks in the middle and lower reaches of the basin was the main source of $Na^+$ and $K^+$ in the river water (*Anshumali, Yadav & Kumar, 2014*). There was a high correlation between $Na^+$, $K^+$, and $Cl^-$ in PC1 (Fig. 3). The $Na^+$ and $K^+$ in the river water were mainly derived from atmospheric precipitation, evaporite and rock weathering, and human activity, while $Cl^-$ was mainly derived from human activity and atmospheric precipitation (*Qin et al., 2018*; *Yan et al., 2022*). After removing $Na^+$, $K^+$, and $Cl^-$ from atmospheric precipitation sources of silicate rock weathering rivers, the remaining $Na^+$, $K^+$ and $Cl^-$ were from anthropogenic sources (*Lü et al., 2018*; *Qin et al., 2018*). It is noteworthy that PC1 also showed a significantly positive loading of $Ca^{2+}$ and $SO_4^{2-}$, indicating $H_2SO_4$–involved carbonate weathering processes in the upstream region (*Barnes & Raymond, 2009*; *Huang et al., 2019*; *Ma et al., 2023*), whereas $Ca^{2+}$, $Mg^{2+}$, and $HCO_3^-$, represented by PC2, were all derived from the weathering of carbonate rocks in the upper reaches of the Qingshuijiang River Basin (*Wu et al., 2023*). $F^-$ and $NO_3^-$ were highly correlated in PC3 and were derived from anthropogenic sources.

### Ion ratio method—reveals the source of the main ions

#### Anthropogenic inputs

When analyzing sources of major ions in water chemistry, high $Cl^-$ concentration in river water can generally be used as an important indicator of human input of domestic sewage (mainly concentrated in urban areas; (*Tang, Jin & Liang, 2021*)). Recent studies have shown that river nitrates from agricultural synthetic fertilizers have higher $NO_3^-$

**Table 2 Factor load of main ions in the Qingshuijiang River Basin.**

| Variable | PC1 | PC2 | PC3 | Communalities |
|---|---|---|---|---|
| $F^-$ | 0.16 | 0.21 | 0.90 | 0.88 |
| $NO_3^-$ | 0.21 | 0.12 | 0.90 | 0.87 |
| $Cl^-$ | 0.92 | 0.26 | 0.08 | 0.91 |
| $SO_4^{2-}$ | 0.86 | 0.30 | 0.35 | 0.95 |
| $Na^+$ | 0.91 | 0.09 | 0.31 | 0.92 |
| $K^+$ | 0.95 | 0.20 | 0.07 | 0.94 |
| $Ca^{2+}$ | 0.74 | 0.60 | 0.22 | 0.95 |
| $Mg^{2+}$ | 0.28 | 0.89 | 0.29 | 0.96 |
| $HCO_3^-$ | 0.22 | 0.96 | 0.10 | 0.96 |
| Eigenvalues | 5.80 | 1.40 | 1.19 | — |
| Variance (%) | 64.41 | 15.45 | 13.25 | — |
| Cumulative (%) | 64.41 | 79.86 | 93.11 | — |

Note:
 Extraction method—principal component analysis; rotation method—Caesar's Normalized Maximum Variance.

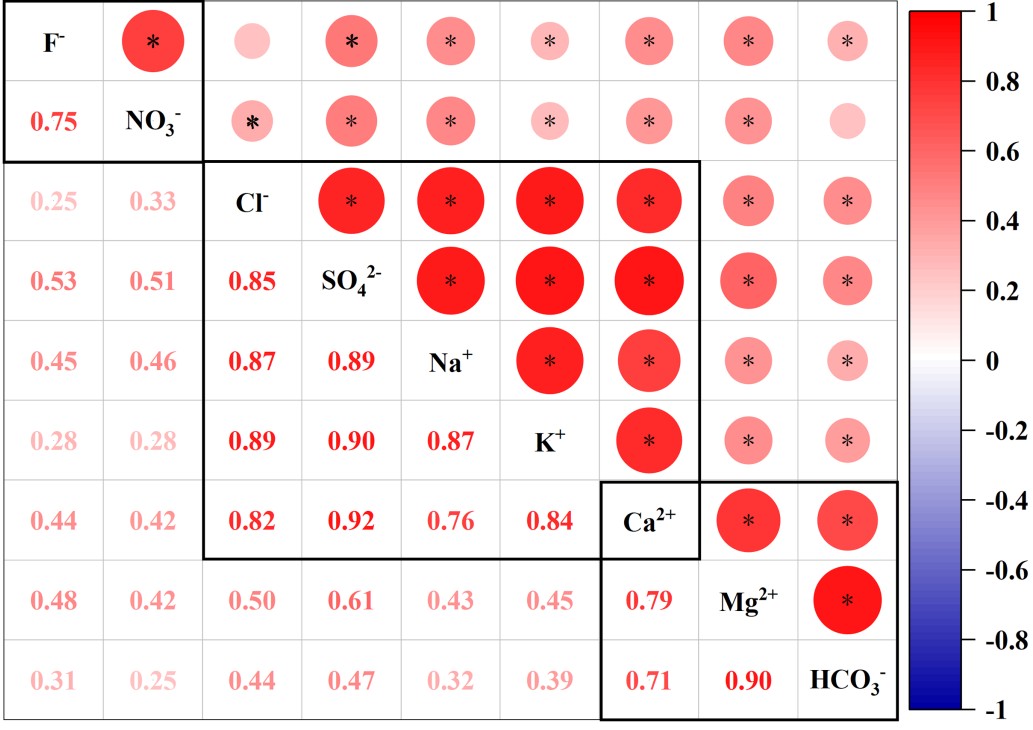

**Figure 3 Pearson correlation matrix of the major ion geochemistry of the Qingshuijiang River.** *
Strong positive correlation coefficients at the 0.001 level (two-tailed).

concentrations and higher $NO_3^-/Cl^-$ ratios, while domestic sewage has lower $NO_3^-/Cl^-$ ratios and higher $Cl^-$ concentrations due to higher organic matter content (*Ge et al., 2021*; *Liu et al., 2021c*; *Yue et al., 2020*). To better explore the changes in the main ion sources in the Qingshuijiang River Basin in the past decade, the main ions were analyzed in the four

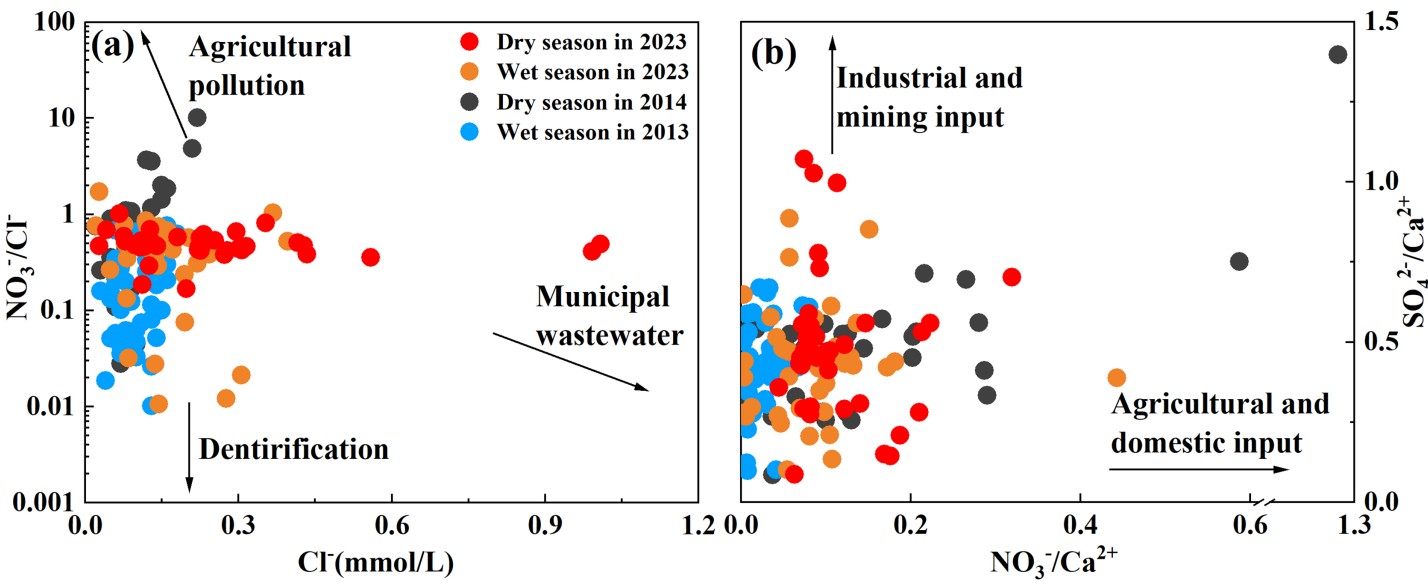

**Figure 4** The relationship between Cl$^-$ and NO$_3^-$/Cl$^-$ ratios (A), NO$_3^-$/Ca$^{2+}$ ratios, and SO$_4^{2-}$/Ca$^{2+}$ ratios (B).

sample periods to identify the sources of the ions. As shown in Fig. 4A, the Cl$^-$ concentration in all water samples was between 0.02–1.00 mmol/L, the NO$_3^-$/Cl$^-$ ratio was between 0.01–10.13, and the coefficient of variation was 86.24% and 160.39%, respectively. This showed that some areas in the basin were affected by urban activities and agricultural activities, with large regional differences in the scope of the effects. Smaller flows were much more vulnerable to pollution than larger flows (*Im et al., 2020*). The comparison of data over the past decade showed that the average content of Cl$^-$ in the Qingshuijiang River Basin doubled from 0.10 to 0.20 mol/L, while the average ratio of NO$_3^-$/Cl$^-$ decreased from 0.74 to 0.49. These results indicate that with the increases seen in population, urbanization, and construction in the Qingshuijiang River Basin, the main source of ions in river water shifted from agricultural input (fertilizer) to municipal wastewater input (*Zheng et al., 2022*). For both urban wastewater and agricultural fertilizer inputs, the impact was greater during the dry season than during the wet season.

High concentrations of SO$_4^{2-}$ in river water are generally derived from the input of mining and industrial production (*Liu & Han, 2020a*), while Ca$^{2+}$ ions are usually derived from rock weathering and are not affected by human factors. Therefore, NO$_3^-$ (representing agricultural and domestic input) and SO$_4^{2-}$ (representing industrial and mining input) were compared with Ca$^{2+}$ to determine the main anthropogenic contributor to solute in the Qingshuijiang River. As shown in Fig. 4, in addition to agricultural activities and urban sewage discharge, industrial activities and mining also impact the solute in the Qingshuijiang River. In the upstream area (Chonganjiang River Basin), which is affected by acid mine drainage, the sulfate in the river water (range 0.50–4.87 mmol/L, average: 1.57 mmol/L) was more than three times the average value (0.49 mmol/L) of the basin (*Li et al., 2024*). The average ratios of SO$_4^{2-}$/Ca$^{2+}$ and NO$_3^-$/Ca$^{2+}$ in the basin were also calculated for the study years. The ratio of SO$_4^{2-}$/Ca$^{2+}$ increased slightly over the

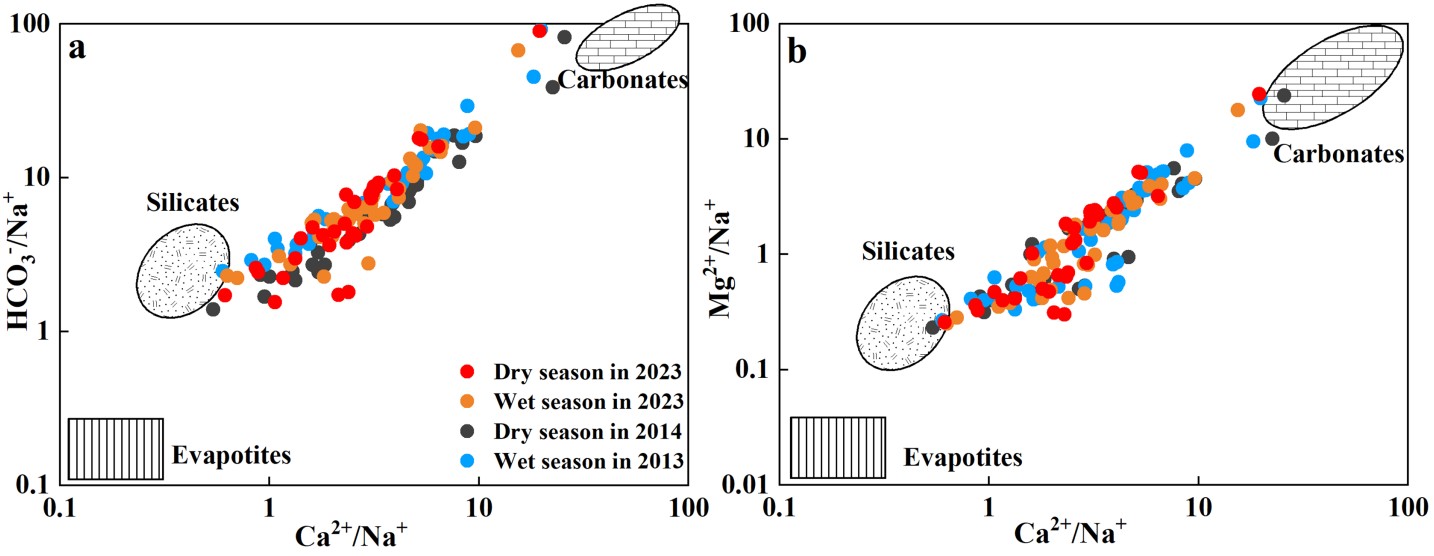

**Figure 5 The relationship between Cl⁻/Na⁺ and HCO₃⁻/Na⁺ ratios (A) and Ca²⁺/Na⁺ and Mg²⁺/Na⁺ ratios (B).**

10-year period, while the ratio of $NO_3^-/Ca^{2+}$ decreased slightly, indicating that some areas in the basin are affected by industrial activities.

### Rock weathering input

Rock weathering includes both carbonate rock weathering and silicate rock weathering, which are the main sources of $Ca^{2+}$, $Mg^{2+}$, $Na^+$, $K^+$, and $HCO_3^-$ in river water (*Herath et al., 2022*; *Tsering et al., 2019*). $Ca^{2+}/Na^+$ and $HCO_3^-/Na^+$ ratios are effective indices for tracing the source of ion weathering (*Liu & Han, 2020b*; *Zheng et al., 2023*). The ratios of $Ca^{2+}/Na^+$, $Mg^{2+}/Na^+$, and $HCO_3^-/Na^+$ in the Qingshuijiang River Basin were dispersed between silicate and carbonate end members (Figs. 5A and 5B), indicating that both carbonate and silicate rock weathering were involved. This reflects the fact that the upstream area is a carbonate rock area and the downstream area is a silicate rock area.

In general, the $Ca^{2+} + Mg^{2+}/(Na^+ + K^+)$ equivalent ratio can be used as an index to distinguish the relative intensity of different types of rock weathering (*Gupta, Nayek & Chakraborty, 2016*; *Setia et al., 2021*). The $Ca^{2+} + Mg^{2+}/(Na^+ + K^+)$ ratio in the Qingshuijiang River Basin ranged from 0.71 to 35.29, with an average of 5.51, which was higher than the world average (2.2) and the Indian average (2.5); (*Setia et al., 2021*), indicating that the chemical composition of the river was more controlled by the lithology of the carbonate rocks in the basin than by the lithology of the silicate rocks. Therefore, a further analysis of the weathering process of carbonate rocks was carried out (Figs. 6A–6C). Most of samples from the Qingshuijiang River Basin were distributed in the $2(Ca^{2+} + Mg^{2+})/HCO_3^-$ concentration ratio (Fig. 6A), indicating that in the process of carbonate rock weathering, $Ca^{2+}$ and $Mg^{2+}$ ions in the river water were more abundant than $HCO_3^-$, and there were other exogenous acids (sulfuric acid or nitric acid) to balance the river water ions (*Gong et al., 2024*; *Li et al., 2023*). Compared with 2013/2014, more exogenous acids were needed in 2023.

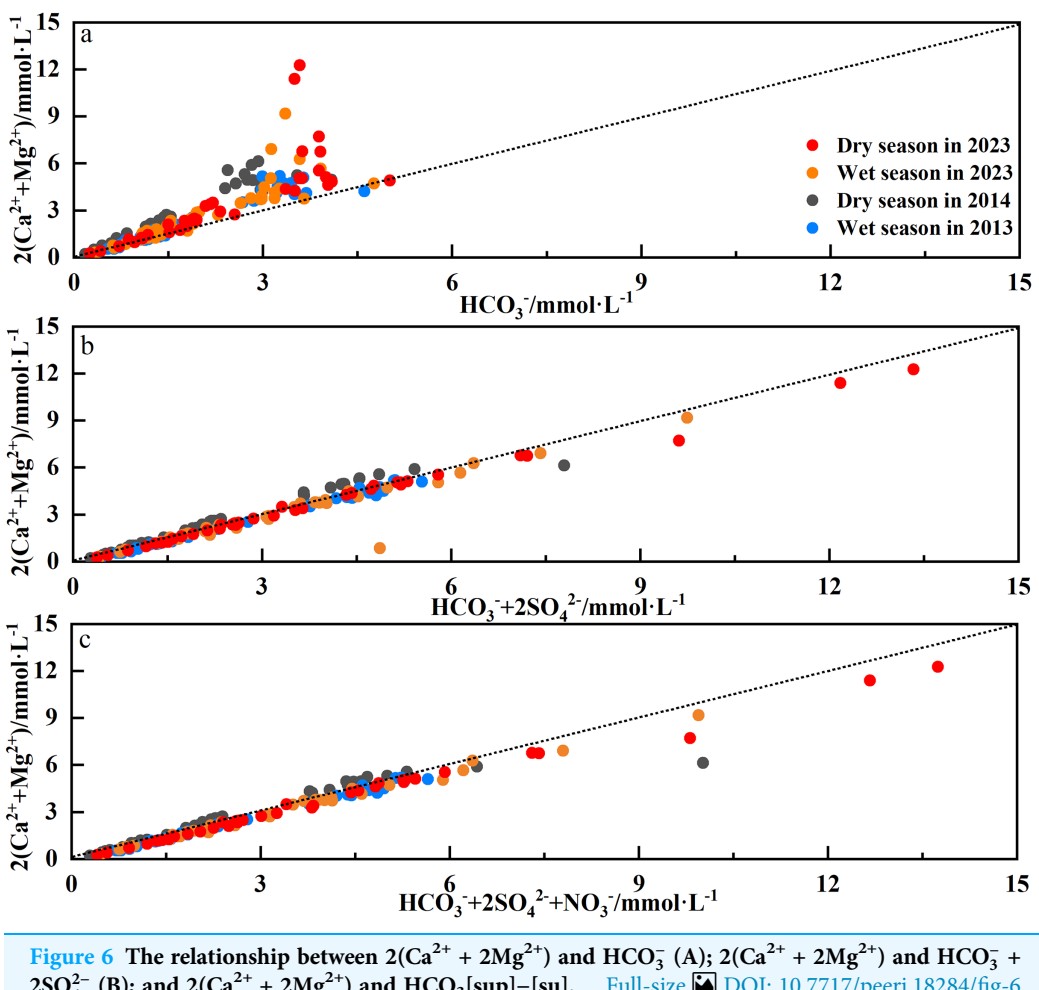

**Figure 6** The relationship between $2(Ca^{2+} + 2Mg^{2+})$ and $HCO_3^-$ (A); $2(Ca^{2+} + 2Mg^{2+})$ and $HCO_3^- + 2SO_4^{2-}$ (B); and $2(Ca^{2+} + 2Mg^{2+})$ and $HCO_3[sup]-[su]$.

In order to further explore the effect of sulfuric acid or nitric acid on weathering, $2(Ca^{2+} + Mg^{2+})$ and $(HCO_3^- + 2SO_4^{2-})$ and $2(Ca^{2+} + Mg^{2+})$ and $(HCO_3^- + 2SO_4^{2-} + NO_3^-)$ were analyzed. As shown in Figs. 6B and 6C, most of the sampling points in the Qingshuijiang River were distributed on the 1:1 contour line on the relationship diagram of $2(Ca^{2+} + Mg^{2+})$ and $(HCO_3^- + 2SO_4^{2-})$, while on the relationship diagram of $2(Ca^{2+} + Mg^{2+})$ and $(HCO_3^- + 2SO_4^{2-} + NO_3^-)$, some sampling points in the Qingshuijiang River fell farther away from the 1:1 contour line, indicating that sulfuric acid was involved in ion balance in the river rather than nitric acid. This result indicates that sulfuric acid might be involved in the weathering of carbonate rocks in the basin (*Li et al., 2008*; *Ma et al., 2023*; *Tang & Han, 2021*). The results of the PCA and CA analyses showed that $Ca^{2+}$ and $SO_4^{2-}$ were in the same component, and the correlation between them was high, which also provides evidence that sulfuric acid is involved in carbonate weathering. The $2SO_4^{2-}$ content involved in the ionic equilibrium of the river water increased from 0.39 to 0.58 mol/L over the 10-year period, suggesting that sulfuric acid played a more significant role in weathering in the clearwater river basin in 2023 than it did a decade prior.

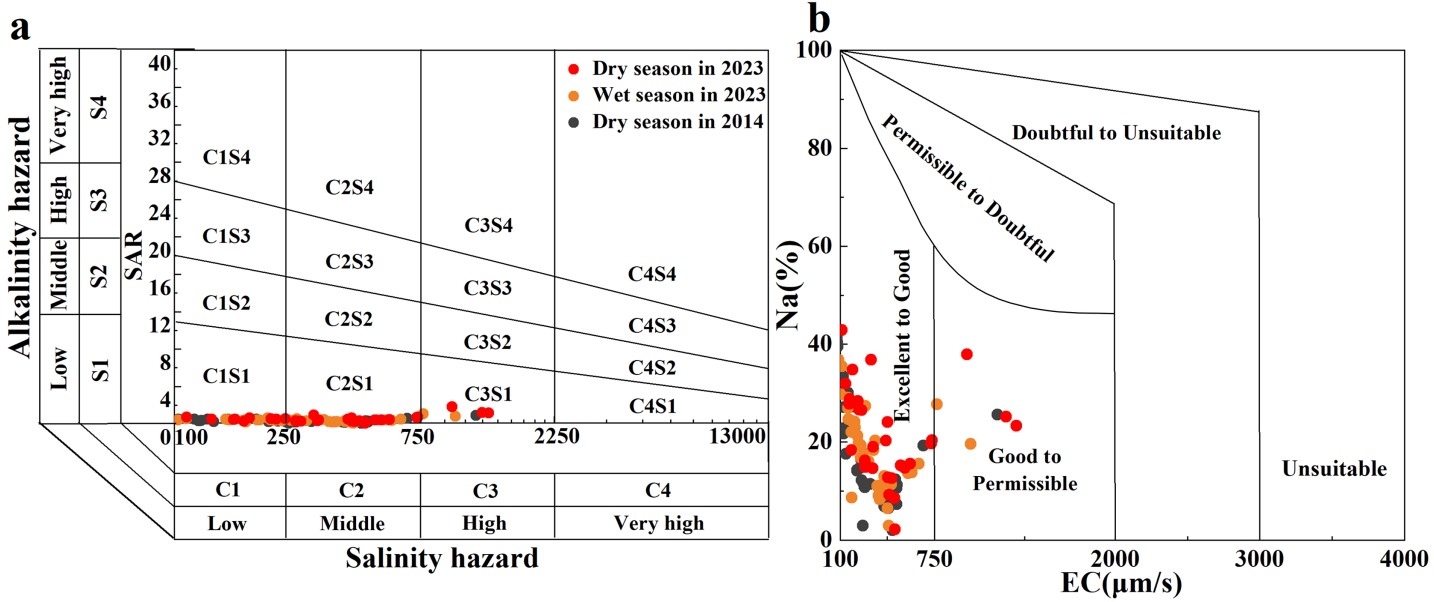

**Figure 7 Salinity and alkalinity evaluation of irrigation water quality: (A) United States Salinity Laboratory (USSL) diagram and (B) Wilcox diagram.**

### Irrigation and guideline-based water quality

The Qingshuijiang River Basin is the main water source for agriculture, industry, and local residents in the Qiandongnan and Qiannan Prefectures of Guizhou Province. The quality of the water in the river basin is closely related to the health of these residents. As summarized in Table 1, the pH values of 82.93% of the sampling points in the Qingshuijiang River Basin were in line with both Chinese and WHO drinking water quality guidelines (6.5–8.5), though the pH values of some sampling points affected by carbonate weathering exceeded 8.5 (*Ge et al., 2021*). Most of the $F^-$, $Cl^-$, $NO_3^-$, and $SO_4^{2-}$ levels in the river water samples were lower than the recommended limits, but $F^-$, $NO_3^-$, and $SO_4^{2-}$ levels at a few of the sampling points exceeded the recommended values. The percentage of sampling points exceeding the recommended values of $F^-$, $NO_3^-$, and $SO_4^{2-}$ were 4.27%, 0.61%, and 2.44%, respectively. These excessive amounts might be related to the phosphorus and fluoride chemical enterprises and coal mining enterprises in the basin (*Tang et al., 2022*; *Van Stempvoort et al., 2023*).

Common indicators for evaluating river irrigation water quality include Na%, SAR, and RSC. Na% and SAR indicators can reflect the Na hazard of soil aggregates affected by irrigation on agricultural land (*Li, Wu & Qian, 2015*). A United States Salinity Laboratory (USSL) diagram and Wilcox diagram were drawn using the EC, SAR, and Na% values of the river water (Figs. 7A and 7B; (*Bishwakarma et al., 2022*)). Most of the Qingshuijiang River water samples were scattered in the C1S1 and C2S1 regions of the USSL map and in the 'excellent' region of the Wilcox map. Only a few sampling points in the tributaries of the Qingshuijiang River were scattered in the C3S1 region of the USSL plot and in the 'good' region of the Wilcox plot. For residual sodium carbonate (RSC), the RSC value of water samples in the basin ranged from −2.55 to 2.56, with an average of 0.56. Overall, the

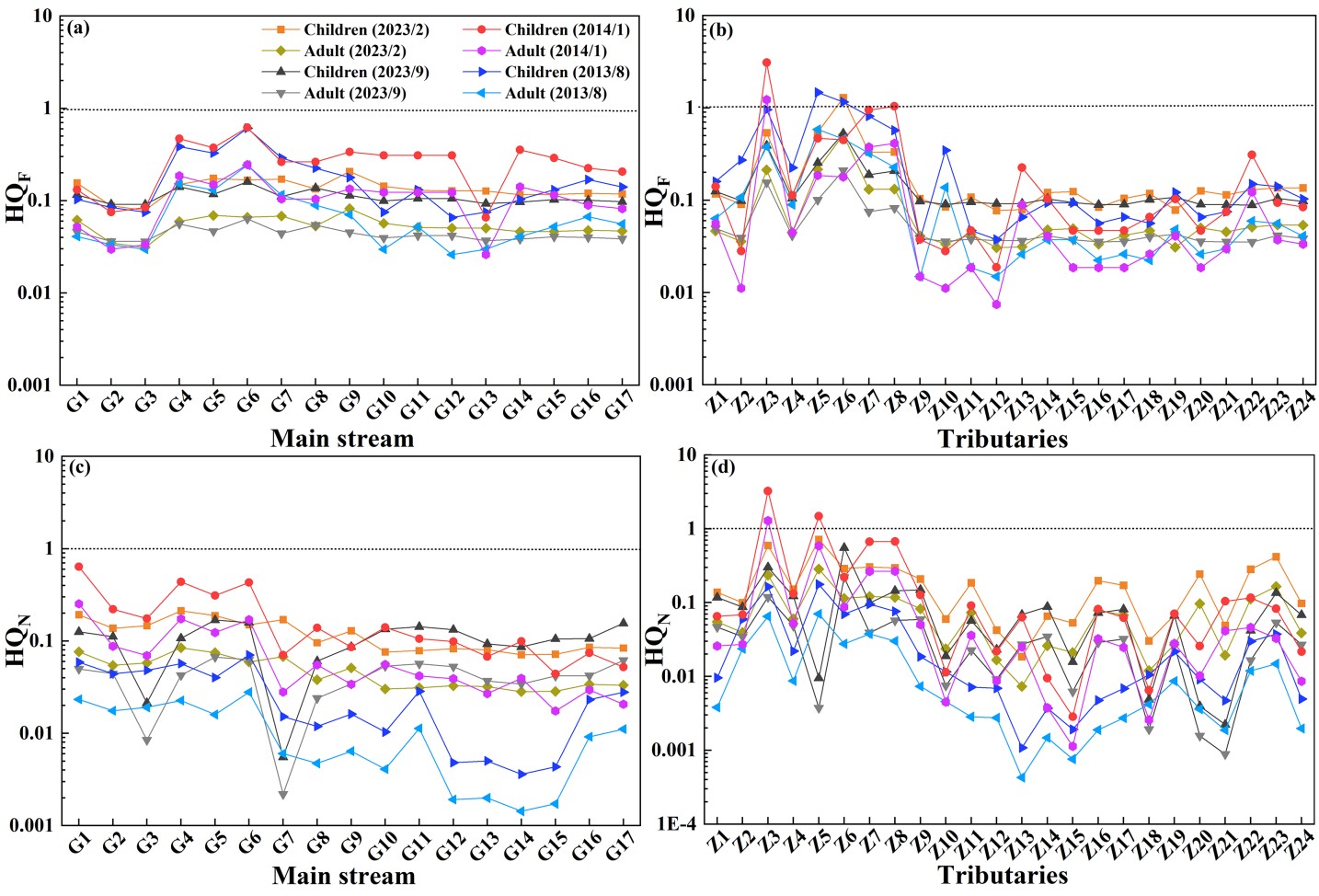

**Figure 8** HQ values of fluoride ($HQ_F$) and nitrate ($HQ_N$) for children and adults in the mainstream (A and C) and (B and D) tributaries of the Qingshuijiang Basin in different sample periods.

results show that the Qingshuijiang River water is not a hazard to the soil when used for agricultural irrigation. However, it is worth noting that the sampling points have slightly shifted toward the direction of poor water quality over the past 10 years, which may be related to the increase in human activity seen in the basin in the past decade. Therefore, the continuous long-term monitoring of the basin is still important.

### Health risk assessment

The Qingshuiiang River Basin is an important source of drinking water for residents near both the Qiandongnan Prefecture and the Qiannan Prefecture. The ingestion of excessive $F^-$ and $NO_3^-$ can cause typical non-carcinogenic hazards, while $SO_4^{2-}$ does not cause health problems (*Liu & Han, 2020a*). Therefore, in this study, $NO_3^-$ and $F^-$ were included in the health risk assessment, and the HQ values of nitrate ($HQ_N$) and fluoride ($HQ_F$) of each sampling point were calculated according to the corresponding concentrations in the river water, and the potential risks to human health were evaluated. As shown in Fig. 8, the HQ values for children in the whole Qingshuijiang River Basin were $HQ_F$ (average: 0.22) > $HQ_N$ (average: 0.14), and those for adults were $HQ_F$ (average: 0.09) > $HQ_N$ (average: 0.06).

These results indicated that the overall health risk of water quality in the mainstream was low, and the health risk of children was higher than that of adults. These results also showed that the potential effect of $F^-$ on health was greater than that of $NO_3^-$. However, the difference between the mainstream and the tributaries was large, and the difference between different tributaries was also large. In some upstream tributaries, the HQ value of some sampling points (Z3, Z5, Z6, and Z8 in Chonganjiang River) exceeded 1, indicating a large health risk. These high values may be related to the large-scale phosphorus chemical base in Fuquan City (*Tang et al., 2022*). The $HQ_F$ and $HQ_N$ values for children in the whole Qingshuijiang River Basin were higher in the dry season (average: 0.24 and 0.21, respectively) than in the wet season (average: 0.19 and 0.06, respectively). In addition, the $HQ_F$ and $HQ_N$ values for children were higher in 2013/2014 (average: 0.27 and 0.14, respectively) than in 2023 (average: 0.15 and 0.13, respectively), indicating that the health risks to children of fluoride and nitrogen in the Qingshuijiang River Basin have declined in the past decade, which may be related to the management of the basin, especially the treatment of key pollution sources. However, it is worth noting that in some areas (Z6) during the dry season, the health risks increased, emphasizing the need for continued environmental management.

## CONCLUSIONS

In this study, the hydrochemistry of surface water in the Qingshuijiang River was investigated during both the wet and dry seasons, 10 years apart. The main sources and 10-year evolution of the main ions in the mainstream and tributaries of the Qingshuijiang River were determined using various statistical methods, including principal component analysis and chemometrics. Rock weathering input (mainly upstream carbonate) was the main source of $Mg^{2+}$, $Ca^{2+}$, and $HCO_3^-$, while $K^+$ and $Na^+$ were affected by a combination of human activity and silicate rock weathering in the middle and lower reaches. Human input was the main source of $SO_4^{2-}$, $NO_3^-$, and $F^-$ ions. In the past 10 years, due to increases in industrialization and population growth in the basin, the concentration of the main ions in the river water has increased significantly, with sulfuric acid now being more involved in the process of rock weathering. Both the water quality assessment and hazard quotient assessment produced good results, indicating that the river water is generally safe for irrigation and drinking, and the health risks are low. However, continuous monitoring of safety is important, especially the risk of excessive $F^-$ in a few tributaries in the basin. This work will help to clarify the hydrochemical characteristics of the Qingshuijiang River Basin under human activities and provide a reference for the sustainable management of the southwest karst river basin.

## ACKNOWLEDGEMENTS

The authors would like to thank Jie Ding, Dong Cai, and Tingting Zhu for their help with sample collection.

### Funding

This work was funded by the High-Level Talent Introduction Program for the Guizhou Institute of Technology (No. 2023GCC083), and the Young Scientific Technical Talents Development Fund of Guizhou Province (No. QJJ[2024]169). The funders had no role in study design, data collection and analysis, decision to publish, or preparation of the manuscript.

### Grant Disclosures

The following grant information was disclosed by the authors:
High-Level Talent Introduction Program for the Guizhou Institute of Technology: 2023GCC083.
Young Scientific Technical Talents Development: QJJ[2024]169.

### Competing Interests

The authors declare that they have no competing interests.

### Author Contributions

- Jiemei Lv conceived and designed the experiments, performed the experiments, analyzed the data, prepared figures and/or tables, authored or reviewed drafts of the article, and approved the final draft.
- Tianhao Yang conceived and designed the experiments, performed the experiments, prepared figures and/or tables, authored or reviewed drafts of the article, and approved the final draft.
- Yanling An conceived and designed the experiments, analyzed the data, prepared figures and/or tables, and approved the final draft.

### Data Availability

The raw measurements are available in the Supplemental File.

### Supplemental Information

Supplemental information for this article can be found online at http://dx.doi.org/10.7717/peerj.18284#supplemental-information.

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
