# Peer review of "Compositions of the major ions, variations in their sources, and a risk assessment of the Qingshuijiang River Basin in Southwest China: a 10-year comparison of hydrochemical measurements"

_PeerJ, doi:10.7717/peerj.18284_

## Round 0.1 · original submission · Major Revisions

Based on the anonymous reviews, your manuscript required major revision before it can be accepted. Please try to submit the revised manuscript and reply within three weeks.

Reviewer 1 ·

Basic reporting

no comment

Experimental design

no comment

Validity of the findings

no comment

Additional comments

The authors evaluated the ionic composition, sources, and health risks of F and nitrate ions in the river waters of the Qingshuijiang River Basin, which is of practical importance. However, several issues must be addressed.
1) There was a discrepancy between the title and content of the article. The title clearly states that the change is over a 10-year period, but the content of the article is only the data from to 2013-2014 and 2023, which does not reflect the 10-year trend but only shows the difference between the current situation and the situation 10 years ago. However, the comparative content of the article regarding the difference is small and cannot reflect the value of the data.
2) In the preface, the authors devoted much space to the practical value of the study but did not reflect the scientific value of the research in this paper. The authors should have clearly stated the scientific problems to be addressed in this study.
3) The sampling section is overly simplistic, and the sampling details are critical for the analysis of the ionic composition. Analyzing the data requires charge balance error analysis of the ionic components.
4) The authors' human health risk is based on the premise that people drink surface water directly; however, what is the source of water for people in the area? If treated with drinking water, what is the significance of the author's health risk evaluation?
5) In the Discussion section, the authors need to be clear about principal component analysis and factor analysis, which are different methods of analysis.
6) In addition, the authors mentioned that " After deducting Na+ and K+ from atmospheric precipitation sources of silicate rock weathering rivers, the remaining Na+, K+ and Cl2 were from anthropogenic sources." The authors must provide a detailed description of the methodology used for the deductions.
7) The authors need to identify the source of SO4. In the principal component analysis, the authors believe that it has the same source as Ca, but in the discussion, the authors believe that SO4 mainly comes from industrial activities.
8) Overall, this study analyzes the ionic composition data in detail based on traditional methods, which has some practical value, but some additions need to be made to the scientific value of the research process of the paper.

Reviewer 2 ·

Basic reporting

good

Experimental design

good

Validity of the findings

good

Additional comments

I have reviewed the manuscript. The authors used principal component analysis (PCA), ion ratio and other methods to study the water chemistry of Qingshuijiang River Basin in the past ten years. And the authors thought it is still necessary to pay attention to the irrigation safety and F exceeding risk of a few tributaries in the basin, and long-term monitoring of the river basin is still essential. I suggest the authors should address the below comments.

1. Line 19, why "daily"?
2. The English should be polished by a fluent English speaker.
3. This study should compare with other's study in the karst area.
4. The format of the writing should be uniform, e.g. line 133
5. The short term should be explained by the first time. e.g. HQ in Line 144, and hazard quotient (HQ) in 153line
6. What is the significant digit after the decimal point, and it shows the precise of measurement .e.g. Line 173
7. What is "、" in English?
8. sufluric acid-driven weathering in Karst area should include relevant references
9. Did aquatic photosysthesis affect DIC and major ions?

---

## Round 0.2 · accepted · Accept

Thank you for addressing all coments from the two anonymous reviewers. We did not received further comments to your manuscript. We agree with the response and the modifications to the manuscript. It can be accepted for publication in PeerJ.